

# Three-dimensional facial development of children with unilateral cleft lip and palate during the first year of life in comparison with normative average faces

Sander Brons[1], Jene W. Meulstee[2], Tom G.J. Loonen[2], Rania M. Nada[3], Mette A.R. Kuijpers[1], Ewald M. Bronkhorst[4], Stefaan J. Bergé[5], Thomas J.J. Maal[2] and Anne Marie Kuijpers-Jagtman[1]

[1] Department of Dentistry, Section of Orthodontics and Craniofacial Biology, Radboud University Medical Centre, Nijmegen, The Netherlands
[2] Department of Oral and Maxillofacial Surgery, Radboudumc 3D Lab, Radboud University Medical Centre, Nijmegen, The Netherlands
[3] Faculty of Dentistry, Kuwait University, Kuwait City, Kuwait
[4] Department of Dentistry, Section of Preventive and Curative Dentistry, Radboud University Medical Centre, Nijmegen, The Netherlands
[5] Department of Oral and Maxillofacial Surgery, Radboud University Medical Centre, Nijmegen, The Netherlands

Corresponding author
Sander Brons,
sander.brons@radboudumc.nl

## ABSTRACT

**Background:** Stereophotogrammetry can be used to study facial morphology in both healthy individuals as well as subjects with orofacial clefts because it shows good reliability, ability to capture images rapidly, archival capabilities, and high resolution, and does not require ionizing radiation. This study aimed to compare the three-dimensional (3D) facial morphology of infants born with unilateral cleft lip and palate (UCLP) with an age-matched normative 3D average face before and after primary closure of the lip and soft palate.

**Methods:** Thirty infants with a non-syndromic complete unilateral cleft lip, alveolus, and palate participated in the study. Three-dimensional images were acquired at 3, 6, 9, and 12 months of age. All subjects were treated according to the primary surgical protocol consisting of surgical closure of the lip and the soft palate at 6 months of age. Three-dimensional images of UCLP patients at 3, 6 (pre-treatment), 9, and 12 months of age were superimposed on normative datasets of average facial morphology using the children's reference frame. Distance maps of the complete 3D facial surface and the nose, upper lip, chin, forehead, and cheek regions were developed.

**Results:** Assessments of the facial morphology of UCLP and control subjects by using color-distance maps showed large differences in the upper lip region at the location of the cleft defect and an asymmetry at the nostrils at 3 and 6 months of age. At 9 months of age, the labial symmetry was completely restored although the tip of the nose towards the unaffected side showed some remnant asymmetry. At 12 months of age, the symmetry of the nose improved, with only some remnant asymmetry noted on both sides of the nasal tip. At all ages, the mandibular and chin regions of the UCLP patients were 2.5–5 mm posterior to those in the average controls.

**Conclusion:** In patients with UCLP deviations from the normative average 3D facial morphology of age-matched control subjects existed for the upper lip, nose, and even the forehead before lip and soft palate closure was performed. Compared to the controls symmetry in the upper lip was restored, and the shape of the upper lip showed less variation after primary lip and soft palate closure. At this early age, retrusion of the soft-tissue mandible and chin, however, seems to be developing already.

## INTRODUCTION

Orofacial clefts (OFCs) are one of the most frequently diagnosed congenital craniofacial malformations (*World Health Organization (WHO), 2000*; *Kadir et al., 2017*). Orofacial clefts lead to several problems such as impairment of facial and dental development, speech and hearing, and facial esthetics. Therefore, affected individuals are liable to suffer stigmatization, social exclusion, and barriers to employment (*World Health Organization (WHO), 2000*). To maximize their potential in adolescent and adult life, affected individuals need the care of an interdisciplinary team of specialists (*Shaw et al., 2001*; *Ness et al., 2015*; *American Cleft Palate Craniofacial Association (APCPA), 2019*).

Treatment of OFCs starts within the first months after birth. Primary treatment of a complete unilateral or bilateral cleft may include presurgical infant orthopedics (plate and/or taping of the lip and/or nasoalveolar molding) and primary surgical correction of the cleft lip and palate. Different techniques and timelines have been described for infant orthopedics and cleft lip closure. The use of numerous different protocols for primary treatment of OFCs across different treatment centers indicates the lack of clear scientific evidence favoring one method over the others (*Shaw et al., 2000*).

Facial morphology is an important outcome variable of cleft treatment, and various methods for assessment of facial morphology have been described in the literature, including direct physical measurements (*Farkas, Hajniš & Posnick, 1993*; *Reddy et al., 2010*), rating of standardized clinical photographs (*Chowdri, Darzi & Ashraf, 1990*; *Halli et al., 2012*), as well as sophisticated three-dimensional (3D) imaging techniques (*Kuijpers et al., 2014*). A systematic review of the literature regarding the reliability and application of 3D facial imaging methods in babies and young children, both unaffected control subjects as well as subjects with OFCs, concluded that stereophotogrammetry is the preferred method due to its millisecond fast image capture, archival capabilities, high resolution, and no exposure to ionizing radiation (*Brons et al., 2012*). However, only a few studies have used (3D) stereophotogrammetry for evaluation of facial growth and treatment outcome in infants with OFCs until 12 months of age. Two studies used linear and angular measurements for evaluation of facial morphology before and after primary lip closure (*Alazzawi et al., 2017*; *Morioka et al., 2018*), and one study also made a comparison with the facial morphology of a control sample (*Mancini et al., 2018*). Furthermore, one study

used superimposition of individual 3D images for evaluation of facial morphology before and after nasoalveolar molding and primary cheilorhinoplasty (*Wu et al., 2016*). One recent study used superimposition of 3D images with generic meshes for evaluation of facial morphology before and after primary cheilorhinoplasty (*Al-Rudainy et al., 2018*). The use of a generic mesh is a recent advancement in technical analysis of 3D facial images. It allows better utilization of the potential of complex longitudinal 3D information than simple linear and angular measurements, removes the need for localization of landmarks and thus reduces operator error, and allows automatic identification of corresponding facial regions by superimposing 3D images with a generic mesh (*Brons et al., 2019*).

Comparison of the facial morphology of individuals with craniofacial malformations and normal controls can be achieved by registration of average faces of each group (*Hammond et al., 2004*). Construction of average faces from 3D images of patients with OFC and controls aged 8–12 years has been reported earlier (*Bugaighis et al., 2014*). To our knowledge, there is no published report on facial growth and treatment outcome in infants with OFCs and controls matched for age during the first year of life using stereophotogrammetry and superimposition with the use of a generic mesh and average faces. Therefore, the aim of this study was to compare the 3D facial morphology of infants born with unilateral cleft lip and palate (UCLP) with the normative average 3D facial morphology of age-matched control subjects before and after primary closure of the lip and soft palate with the application of a generic mesh.

## MATERIALS AND METHODS

### Ethical approval and informed consent

The medical ethics commission of the institution in which the study was conducted approved the study protocol (Commissie Mensgebonden Onderzoek regio Arnhem-Nijmegen #2007/163). The study was performed in accordance with the 1964 Declaration of Helsinki and its subsequent amendments. Written informed consent from the subjects' parents was obtained prior to their inclusion in the study.

### Subjects

The patients in this study were children with non-syndromic complete UCLP. The subjects in this study are part of a prospective longitudinal 3D study on facial growth from the age of 3 months to 6 years (*Brons et al., 2019*). Data for patients with UCLP were collected within the first month after birth at the Cleft Palate Craniofacial Unit of the Radboud University Medical Centre in Nijmegen, the Netherlands, between September 2008 and December 2011. The inclusion criteria were non-syndromic complete UCLP and age ≤ 3 months at the time of study entry. Only infants born at term (38+ weeks) to parents who were both Caucasians were included. Exclusion criteria were congenital malformations other than UCLP and the presence of soft-tissue bands. A total of 30 patients with UCLP were enrolled in the study.

The control group was recruited before the age of 3 months at the Maternity Clinic of the Radboud University Medical Centre and Regional Health Services (GGD Gelderland-Zuid) between April 2007 and September 2010. Inclusion criteria for the

controls were born at term (38+ weeks) and both parents Caucasian. Exclusion criteria for the controls were occurrence of oral clefts in the first-, second-, or third-degree relatives. Fifty controls were enrolled in the study. In a previous study, normative average faces at 3, 6, 9, and 12 months of age were developed (*Brons et al., 2019*).

## Treatment protocol

Since 2008, the primary surgery protocol at the Nijmegen Cleft Palate Craniofacial Unit consists of surgical closure of the lip and the soft palate at 6 months of age. The lip is closed by a modified Millard technique, and closure of the soft palate is done according to a modified von Langenbeck technique. The hard palate is closed at 3 years of age, and alveolar bone grafting is performed at 9–11 years of age before eruption of the permanent canine.

## 3D image acquisition

Acquisition of 3D images was done within a period of −21 and +21 days around the age of 3, 6, 9, and 12 months. Image acquisition was performed in a designated 3D imaging room with no windows and a consistent amount of ambient lighting. The 3dMDfacial System (3dMD Ltd., Atlanta, GA, USA) with a 2-pod configuration was used for frontal facial image acquisition including both ears. The distance between the infants and cameras was one meter. The camera was calibrated on a daily basis. The 3D images consisted of approximately 20,000 points and the texture map was eight megapixel. Three-dimensional images of the subjects' faces were acquired at 3, 6 (pre-treatment), 9, and 12 months of age at the 3D Lab (Nijmegen, Netherlands) by trained photographers. The duration of the image capture was 1.5 milliseconds. On each occasion, approximately three images were obtained within 10 minutes. Image quality was visually assessed immediately after acquisition by the photographer for completeness of 3D image data and a neutral facial expression using 3dMDpatient V4.0 software.

## Selection of eligible 3D images

High-quality 3D images of the face at rest from the entire sample at 3, 6, 9, and 12 months of age were selected. Inclusion criteria were (1) a neutral facial expression with the eyes open and the lips lightly opposed without straining, (2) orientation of the face in the natural head position, (3) no data holes in the facial region medial to the ears, caudal to the hairline, and cranial to the menton, (4) correct 3D image construction, and (5) no presence of lip tape or a nasal feeding tube. The reliability of the selection procedure was very good, as shown by a kappa value of 0.90 (*Brons et al., 2018*).

## 3D image processing

The selected 3D images were exported from the 3dMDpatient 4.0 software as wavefront object (.obj) files with texture. Next, the 3D images were imported into Maxilim version 2.3.0.3 (Nobel-Biocare, Mechelen, Belgium). The children's reference frame described previously was used to align all 3D images in the correct position and orientation (*Brons et al., 2013*). Third, 3D images of the right-sided UCLP subjects were mirrored on the mid-sagittal plane to obtain 3D images of left-sided UCLP only. Next, remeshing

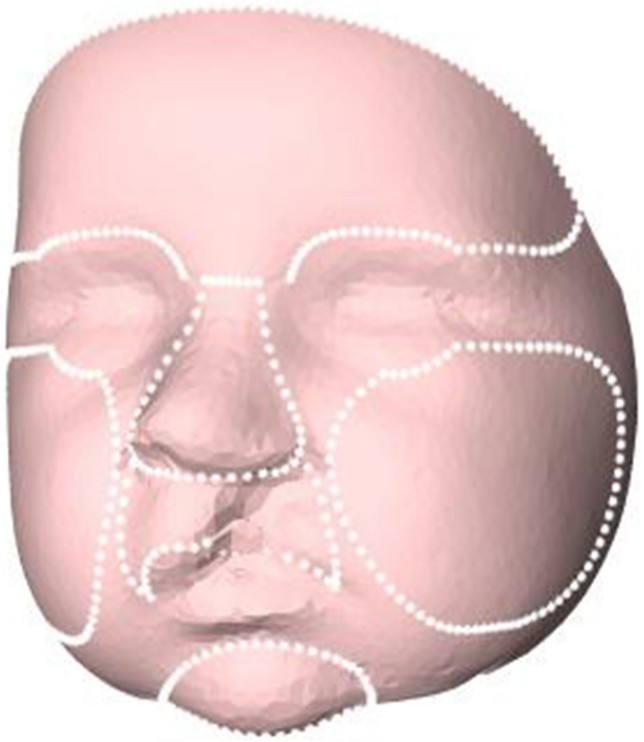

**Figure 1 Selected regions for evaluation of facial growth: total facial surface, nose, upper lip, chin, forehead, and cheeks.**

of the 3D images was performed in Meshmixer software (Autodesk) to obtain a uniform mesh density of vertices which were 1.5 mm apart from each other. Matlab (MathWorks, MA, USA) was then used to automatically annotate the left and right pupil, the pronasale, and the left and right exostomion on the aligned 3D images. The landmarks were indicated on the 2D texture files automatically with a cascaded convolutional network trained by *Zhang et al. (2016)* and transferred to the 3D images.

In the next step, the 3D images were cropped in order to remove excess data of the subjects' head and select only the face. A general face template was scaled to every individual 3D image by a Procrustes transformation based on the landmarks of the left and right pupil, the pronasale, and the left and right exostomion. After the template was scaled to the individual subjects, the outer boundary of the scaled template was used to crop the 3D images and to remove excess data such as hair, ears, and the neck. Then, the Coherent Point Drift algorithm was used for non-rigid deformation of the general face template to the mesh of the 3D images (*Sang, Zhang & Yu, 2013*). After this non-rigid transformation of the general face template, resampling by a ray casting algorithm was performed to create a uniform mesh pattern for all subjects with the same number of vertices (*Meulstee et al., 2017*). From these uniformly resampled 3D images, average faces were created for 3, 6, 9, and 12 months of age. Next, on the general face template (step 5), the regions of the forehead, nose, cheeks, oral region, and chin were selected manually once (Fig. 1) (MLF Hol, JW Meulstee, JM Merks, T Alderliesten, SJ Bergé, AG Becking, LE Smeele, P Hammond, TJJ Maal, 2017, unpublished data). The selected regions

**Table 1 Selection process for the eligible 3D images of UCLP subjects.**

| Age | 3 months | 6 months | 9 months | 12 months | Total |
|---|---|---|---|---|---|
| Children in the database ($n = 30$) | 27 | 29 | 25 | 14 | |
| 3D images in the database | 70 | 71 | 57 | 41 | 239 |
| Excluded images (did not meet inclusion criteria) | 45 | 35 | 30 | 21 | 131 |
| Exclusion of duplicate images | 6 | 11 | 10 | 7 | 34 |
| Exclusion of images with missing corresponding images | 6 | 5 | 1 | 2 | 14 |
| Included children ($n = 20$) | 13 | 20 | 16 | 11 | |

Note:
  UCLP, unilateral cleft lip, alveolus, and palate.

were directly transferred to the individual 3D facial images for 3, 6, 9, and 12 months of age. Finally, all individual 3D images of UCLP patients at 3, 6, 9, and 12 months of age were superimposed on the corresponding normative age-matched average faces based on the children's reference frame (*Brons et al., 2019*). Distance maps of the 3D complete facial surface and of the nose, upper lip, chin, forehead, and cheeks were developed, and comparisons were made of distance kits for intervals of 3–6 months, 6–9 months, and 9–12 months of age.

### Statistical analysis

Color-distance maps are presented for visual assessment of the variations in facial morphology of UCLP patients vs. control subjects. Distance maps of UCLP subjects superimposed on the corresponding normative average age-matched faces were quantified in terms of the mean distance and standard deviation for the face and its regions (nose, oral region, chin, forehead, and cheeks). *T*-tests were applied to assess the differences in the mean distances and mean standard deviations of the distances at intervals of 3–6 months, 6–9 months, and 9–12 months of age for the full face and selected facial regions, with significance set at $p < 0.05$. A 95% confidence interval was applied when interpreting the reliability of the results.

## RESULTS

### Image selection

In total, 239 3D images of 30 UCLP subjects at four different time-points were available. After the selection process, 131 images were excluded because they did not meet the inclusion criteria. From the remaining 108 3D images, 34 3D images were excluded since they were duplicate images of the same patient, and 14 3D images were excluded because the corresponding image before or after primary lip closure was missing. Finally, 60 single 3D images of 20 individual patients at 3, 6, 9, and 12 months of age were included (Table 1).

### Facial morphology of UCLP vs. control

The average faces of UCLP subjects at 3, 6, 9, and 12 months of age are presented in Fig. 2, with those at 9 and 12 months representing the post-operative state. Average faces of control subjects at similar ages were published in a previous study (*Brons et al., 2019*). Visual assessment of the facial morphology of UCLP vs. control subjects is presented in

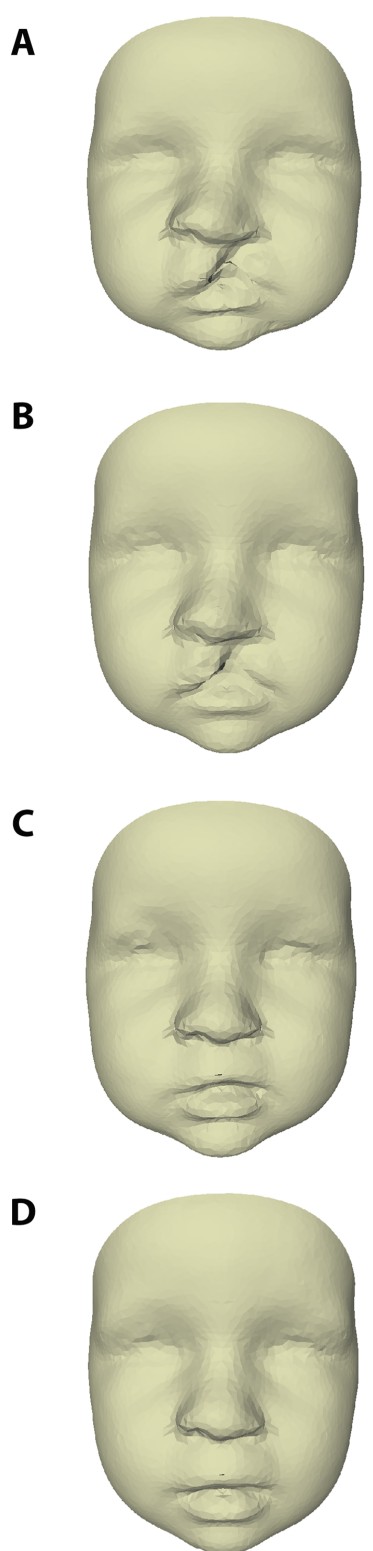

**Figure 2** Average faces of UCLP subjects at age (A) 3, (B) 6, (C) 9, and (D) 12 months of age.

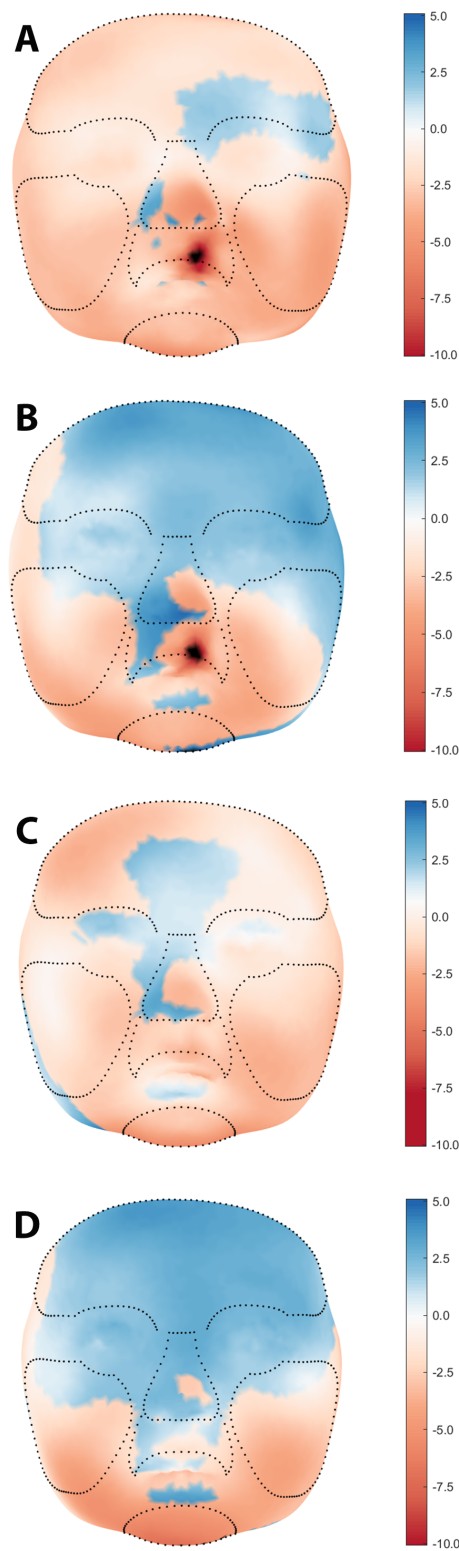

**Figure 3** Visual assessment of average facial morphology of UCLP vs. controls at age (A) 3, (B) 6, (C) 9, and (D) 12 months in color distance map (colour scale red – blue = −10.0 mm – + 5.0 mm).

color-distance maps in Fig. 3. At 3 months of age, the least intersurface distance between UCLP and control facial morphology was noted in the region of the eyes. The intersurface distance increased towards the unoperated cleft lip by over −10 mm, indicating that the average face of the UCLP subjects in this area was posterior to the average face of the control subjects. Asymmetry at the region of the nose was visible with the nostril at the affected side at −7 mm compared to the average control face and that at the unaffected side five mm anterior to the location of the corresponding nostril on the average control face. At 6 months of age, the upper facial half was anterior to the average control face and nasal as well as labial asymmetry had markedly increased. At 9 months of age, the symmetry of the upper lip was completely restored despite the upper lip being an average of four mm posterior compared to the control. Nasal asymmetry, however, was not restored at this age, with the tip of the nose remaining towards the unaffected side and the left and right nostrils at +2.5 mm and −2.5 mm, respectively, compared to the average control face. At 12 months of age, the upper facial half was anterior to the average control face and the symmetry of the nose improved, but asymmetry is still present with only a remnant asymmetry of two mm on both sides of the nasal tip. Asymmetry of the upper lip reoccurred with the unaffected side being one mm anterior to the control. At all ages, the mandibular region and chin region of the UCLP subjects were posterior to the average controls by 2.5–5 mm.

Table 2 presents the mean intersurface distances between the full face and defined regions (nose, upper lip, chin, forehead, and cheeks) of UCLP subjects relative to the average control faces for the same age. In general, the mean distance between the cleft and control was most negative at 3 months of age for the full face and all facial regions, ranging from −3.8 mm to −0.6 mm, meaning that the UCLP subjects on average showed retrusive facial dimensions in an anterior-posterior dimension compared to the average control face of the same age. The standard deviation of the mean distance for the upper lip decreased from 4.1 and 5.0 mm at ages 3 and 6 months, respectively, to 2.5 and 2.0 mm at 9 and 12 months respectively, indicating a decrease in the variation of the intersurface distance between UCLP patients and controls due to the primary operation.

## Facial morphology at age intervals

Table 3 shows the increments in the intersurface distances (mean, sd, $p$-values, and 95% CI) between UCLP subjects' faces relative to the average face of the age-matched control subjects for three age intervals (3–6, 6–9, and 9–12 months of age). In general, the mean increments were negative from 6 to 9 months of age, indicating that during that time period, facial growth in UCLP subjects was smaller than that in the averaged control faces.

Significant differences were found from 3 to 6 months of age for increments in the full face ($p = 0.01$), nose ($p = 0.02$), and cheeks ($p = 0.03$), indicating significantly greater growth in these regions in the UCLP subjects. The mean increment of the standard deviation of the upper lip became significantly smaller ($p = 0.01$) from 6 to 9 months of age, indicating that the shape of the upper lip shows less variation between UCLP subjects and controls at 9 months of age compared to 6 months. Moreover, a significant difference was found from 9 to 12 months of age for the mean increment of the forehead ($p = 0.04$), indicating significantly more growth of the forehead in the UCLP subjects.

**Table 2 Mean distances and standard deviations (in mm) of the full face and regions of the nose, upper lip, chin, forehead, and cheeks of UCLP subjects relative to the age-matched average control face.**

| Region | N | Age (months) | Mean (mm) | Std. (mm) |
|---|---|---|---|---|
| Full face | 13 | 3 | −2.5 | 5.5 |
| | 20 | 6 | 0.0 | 5.2 |
| | 16 | 9 | −0.9 | 4.5 |
| | 11 | 12 | 0.5 | 4.2 |
| Nose | 13 | 3 | −1.3 | 3.7 |
| | 20 | 6 | 0.9 | 3.7 |
| | 16 | 9 | −0.1 | 3.1 |
| | 11 | 12 | 1.8 | 2.0 |
| Upper lip | 13 | 3 | −3.7 | 4.1 |
| | 20 | 6 | −2.2 | 5.0 |
| | 16 | 9 | −2.2 | 2.5 |
| | 11 | 12 | 0.2 | 2.0 |
| Chin | 13 | 3 | −4.2 | 5.9 |
| | 20 | 6 | −1.0 | 4.0 |
| | 16 | 9 | −1.8 | 2.8 |
| | 11 | 12 | −2.2 | 4.8 |
| Forehead | 13 | 3 | −0.6 | 2.8 |
| | 20 | 6 | 1.8 | 3.2 |
| | 16 | 9 | −0.2 | 3.1 |
| | 11 | 12 | 3.0 | 2.6 |
| Cheeks | 13 | 3 | −3.8 | 4.9 |
| | 20 | 6 | −1.2 | 4.1 |
| | 16 | 9 | −1.6 | 3.8 |
| | 11 | 12 | −1.4 | 3.3 |

**Note:**
UCLP, unilateral cleft lip, alveolus, and palate.

Finally, the mean increment of the standard deviation of the nose became significantly smaller ($p = 0.02$) from 9 to 12 months of age, indicating that the shape of the nose shows less variation between UCLP subjects and controls.

## DISCUSSION

Three-dimensional assessments of average facial morphology during the first year of life in children with UCLP compared to non-cleft age-matched peers have not been described so far in literature. However, there is one study in which the degree of 3D facial asymmetry in infants with and without unilateral cleft lip and/or palate was compared to age-matched controls during the first year of life (*Hood et al., 2003*). In the present study, facial morphology was evaluated from 3 to 12 months of age, including primary closure of the lip and soft palate at 6 months of age, by using stereophotogrammetry. At 3 and 6 months of age, color-distance maps have shown large differences in facial morphology of UCLP and control subjects in the upper lip region at the location of the cleft defect and an

**Table 3** *T*-test for equality of increments (mean and std.) between UCLP subjects relative to the average age-matched face of control subjects (*p*-value) for the three age intervals.

| Region | Age interval (months) | Mean increment mean (mm) | 95% CI | *p*-value (mean) | Mean increment std. (mm) | 95% CI | *p*-value (std.) |
|---|---|---|---|---|---|---|---|
| Full face | 3–6 | 2.5 | 0.8–4.2 | 0.01[a] | −0.3 | −1.4–0.8 | 0.55 |
| | 6–9 | −1.0 | −2.7–0.8 | 0.27 | −0.7 | −1.5–0.1 | 0.09 |
| | 9–12 | 1.4 | −1.1–4.0 | 0.26 | −0.2 | −1.3–0.8 | 0.65 |
| Nose | 3–6 | 2.2 | 0.4–4.0 | 0.02[a] | 0.0 | −1.1–1.1 | 0.99 |
| | 6–9 | −1.0 | −2.7–0.8 | 0.27 | −0.7 | −1.7–0.3 | 0.15 |
| | 9–12 | 1.9 | −0.5–4.3 | 0.11 | −1.0 | −1.9 to −0.2 | 0.02[c] |
| Upper lip | 3–6 | 1.5 | −0.9–4.0 | 0.21 | 0.9 | −0.3–2.1 | 0.15 |
| | 6–9 | 0.0 | −2.3–2.3 | 0.99 | −2.5 | −3.5 to −1.5 | 0.01[b] |
| | 9–12 | 2.4 | −0.7–5.5 | 0.13 | −0.5 | −1.2–0.2 | 0.15 |
| Chin | 3–6 | 3.2 | −0.4–6.8 | 0.08 | −1.9 | −5.4–1.6 | 0.26 |
| | 6–9 | −0.7 | −4.6–3.1 | 0.70 | −1.2 | −2.9–0.5 | 0.17 |
| | 9–12 | −0.4 | −4.7–3.8 | 0.84 | 1.9 | −0.4–4.3 | 0.10 |
| Forehead | 3–6 | 2.4 | −0.6–5.4 | 0.11 | 0.4 | −0.6–1.4 | 0.44 |
| | 6–9 | −2.0 | −4.9–0.9 | 0.17 | −0.1 | −1.1–0.9 | 0.87 |
| | 9–12 | 3.3 | 0.1–6.4 | 0.04[c] | −0.4 | −1.5–0.6 | 0.42 |
| Cheeks | 3–6 | 2.6 | 0.2–5.0 | 0.03[a] | −0.8 | −2.2–0.5 | 0.21 |
| | 6–9 | −0.4 | −2.8–2.0 | 0.75 | −0.3 | −1.3–0.6 | 0.50 |
| | 9–12 | 0.1 | −3.1–3.4 | 0.93 | −0.4 | −1.5–0.7 | 0.42 |

**Notes:**
UCLP, unilateral cleft lip, alveolus, and palate; Std, standard deviation.
[a] Significant differences in the age interval of 3–6 months in the mean difference for the full face, nose, and cheeks.
[b] Significant differences in the age interval of 6–9 months in the mean difference for the standard deviation of the upper lip and nose.
[c] Significant differences in the age interval of 9–12 months in the mean difference for the forehead and the standard deviation of the nose.

asymmetry at the nostrils. At interval 3–6 months of age, significant increments were found in the face of UCLP subjects at the regions of the full face, nose, and cheeks. This may be explained by a kind of catch-up growth at age 6 months for these regions, being relatively underdeveloped at age 3 months. At 9 months of age, the labial symmetry was completely restored although the tip of the nose towards the unaffected side showed some remnant asymmetry. At interval 6–9 months of age the trend seems to be that the full face and facial regions grew less in the average UCLP face compared to the average control face. This might be due to the impact of the surgical intervention between 6 and 9 months of age. At 12 months of age, the symmetry of the nose improved, with only some remnant asymmetry noted on both sides of the nasal tip. At all ages, the mandibular region and chin region of the UCLP patients were 2.5–5 mm posterior to those in the average controls. Hood et al. found improvement in the symmetry of the face and nose of the UCLP patients after primary surgery of the lip and nose, however no improvement in symmetry of the lip was found at this interval. Furthermore, they found improvement of symmetry of the lip at ages 6–12 months, however no further improvement of symmetry of the face and nose was found. This is partially in contrast to the results of the present study in which an improvement of symmetry of the lip post-surgery and an improvement of nasal symmetry was found from ages 6 to 12 months. Comparing the results of both studies is challenging

due to differences in the timing of the applied surgical protocols as well as to differences in the methods of analyzing the 3D images.

After reviewing the literature on longitudinal methods for evaluation of facial growth and treatment outcomes in children, we found no method for longitudinal evaluation of 3D facial images of growing individuals. Iterative closest point registration (ICP) is used in non-growing individuals and cross-sectional studies but this method is not appropriate for longitudinal analysis. Iterative closest point has the effect of regression to the mean which is likely to mask any regional differences in individual growth patterns. In the present study, the children's reference frame was used to superimpose 3D images on the average $z$-coordinate of the right and left preauricular points. In this approach, the use of the children's reference frame resembles the traditional and biological registration of lateral cephalograms on the sella turcica of the cranial base. Comparisons of the outcome of our study to other studies are challenging due to variations in the measurement and superimposition techniques. We recommend the use of the children's reference frame for cross-sectional and longitudinal evaluations of changes due to growth or treatment because this is more informative than simplifying facial dimensions into inter-landmark distances and angles. Registration on the children's reference frame combines changes in both shape and size. Placement of the children's reference frame is a reliable technique with a relatively small intra-observer error of 0.40 mm and an interobserver error of 0.51 mm (*Brons et al., 2013*) and this error seems to be clinically insignificant.

The reproducibility of the manual identification of the facial regions was not evaluated because this was performed only once on the general face template. In turn, after mapping of the general face template to the individual 3D facial images, the manual selected facial regions were automatically transferred from the general face template to every individual 3D facial image. We used the same general face template for both the controls and the UCLP patients. Therefore, intra- and inter-study reproducibility of the selection of facial regions is 100% in our case. Manual selection of facial regions is likely to introduce a reproducibility error if another general face template is developed with another manual selection of facial regions. However, the boundaries of the facial regions have little influence on the interpretation of the color-distance maps.

The use of a generic mesh is a recent advancement in technical analysis of 3D facial images and allows greater utilization of the potential of complex longitudinal 3D information compared to simple linear and angular measurements, removes the need for localization of landmarks and reduces operator error, and finally allows automatic identification of corresponding facial regions.

Facial morphology is influenced by variations in facial expressions, even when stereophotogrammetric images with a neutral facial expression are selected. The influence of involuntary facial expressions has shown to range from 0.38 to 0.88 mm in the full facial surface, but the influence is higher in the nasolabial area, especially in UCLP subjects, with mean distances ranging from 0.72 to 0.93 mm at 3–12 months of age (*Brons et al., 2018*). This variation should be taken into account when interpreting the facial

morphology of individuals with craniofacial malformations compared to controls of the same age or longitudinal changes of facial morphology within the same group.

We were able to quantify deviations from the normal facial anatomy in the regions of the upper lip, nose, and even the forehead at 3 and 6 months of age before lip and soft palate closure was performed. The results confirm the presence of intrinsic as well as functional/adaptive differences from birth on between individuals with and without UCLP as already proposed by earlier researchers (*Ross, 1987*; *Markus, Delaire & Smith, 1992*; *Friede, 1998*). *Markus, Delaire & Smith (1992)* and *Friede (1998)* described the morpho-functional anatomy of the nasolabial muscles—the anterior facial muscle chains—in patients with a complete unilateral cleft. The external morphology of our 3D average cleft faces at 3 and 6 months remarkably resembles the anatomy of the affected facial musculature. After primary closure of the lip and soft palate, the symmetry in the upper lip is almost completely restored and the shape of the upper lip shows less variation between UCLP subjects and controls, despite the upper lip being retrusive to the upper lip of the average control face. This indicates that it is possible to surgically restore symmetry of the upper lip, but at this early age, maxillary soft-tissue retrusion seems already developing. Furthermore, the symmetry of the nose improved at 9 months of age and continued to improve until 12 months of age. Longitudinal studies using stereophotogrammetry following patients and controls during childhood are further needed to obtain better insights into soft-tissue changes after the first year of life as facial growth and surgical interventions may introduce improvements in certain areas of the face but may also be responsible for deteriorations.

Stereophotogrammetry has the potential to serve as a method for early detection of facial growth disturbances related to certain surgical treatments as well as soft-tissue changes related to presurgical nasoalveolar molding (PNAM). It would be interesting to compare the effect of the latter therapy on the nasolabial area, as the assumed effect of PNAM especially concerns the shape of the nose. Stereophotogrammetry is a non-invasive and rapid imaging technique, although acquisition of high-quality images is dependent on a highly experienced operator. A stereophotogrammetric setup is expensive, but it provides greater value than conventional two-dimensional photography by allowing quantification and localization of deviations from normal facial development at an early age, as shown in this study. This enables early identification of treatment protocols that are potentially detrimental for the development of normal facial morphology and can be subsequently abandoned earlier. The routine use of stereophotogrammetry in cleft centers is recommended in order to increase the sample size in multi-center studies and for comparison of the outcomes of various cleft treatment protocols. The recent development of handheld 3D cameras will stimulate the use of stereophotogrammetry in the clinic due to its reduced costs and increased mobility (*Kim et al., 2018*).

With stereophotogrammetry, it is possible to detect the static morphology of the face and quantify differences between multiple 3D faces in detail. However, the normal morphology can only partially fulfill the desired goal of a normal facial appearance and esthetics. Normal facial dynamics is another crucial factor for perception of normal facial appearance and esthetics. Therefore, future research should also focus on other

standardized facial expressions besides a relaxed neutral expression, like maximal smile and pouting of the lips.

There are limitations to our study. From our longitudinal prospective 3D facial imaging database of 30 non-syndromic Caucasian UCLP subjects, 11–20 high-quality 3D images were included per age. Initial sample size was lower than expected due to difficulty for the parents to attend at every desired 3D imaging window of time. Second, despite capturing multiple 3D images per occasion many 3D images had to be excluded due to not meeting the inclusion criteria. This is inherent with 3D facial imaging in infants as these subjects are unable to cooperate with the ideal image capture procedure. We did not differentiate between boys and girls in order to obtain a sufficient sample size. This seems to be acceptable since sexual dimorphism in the soft tissue of the face in babies has not been demonstrated in literature, while on the other hand sexual dimorphism in adolescents and adults has been demonstrated (*Bugaighis et al., 2014*). One study in 3-month-old babies reported differences for some facial dimensions, but these differences were related to differences in weight between subjects rather than sexual dimorphism (*Kesterke et al., 2016*). Several studies have demonstrated the presence of craniofacial sexual dimorphism in older children. One study reported sexual dimorphism in the measurements of the cranial vault width and length and facial height in children at 4.7 years of age (*Gaži-Čoklica et al., 1997*). Ferrario demonstrated differences in soft-tissue facial dimensions between boys and girls as young as 6 years of age (*Ferrario et al., 1999*). White found a sex difference of 1–2 mm in larger facial measurements such as the face height and ear-to-chin distance in infants 3 months of age, but this correlated with body measurements. There were no sex differences in the nose/upper lip width ratios or angular measurements, indicating there may be little sex difference in shape (*White et al., 2004*).

A limitation of this study is the relatively large time window in which 3D images were acquired, namely +/− 3 weeks around the exact age of 3, 6, 9, and 12 months. The relatively large time window was necessary to obtain a sufficient sample size. A difference of six weeks at most at these young ages will certainly influence the study outcome. It is recommended that future studies aim to limit the time window in which 3D images are made. Furthermore, future studies may need to acquire 3D images every month, or even every week, to develop accurate growth models for faces. The results of this study are not applicable to other cleft subtypes. Future studies should focus on comparison of normative and UCLP average faces with the developing facial morphology of other cleft subtypes such as unilateral cleft lip and alveolus and bilateral cleft lip, alveolus, and palate.

Another limitation of the study is the inability to distinguish between the effect of intrinsic growth and the effect of treatment in UCLP subjects. To differentiate between these two factors we would need an additional control group in our study of untreated UCLP subjects. However, for obvious ethical reasons it will never be possible to withhold surgical treatment from a group of newborns with clefts.

## CONCLUSIONS

In patients with UCLP deviations from the normative average 3D facial morphology of age-matched control subjects existed for the upper lip, nose, and even the forehead before

lip and soft palate closure was performed. Compared to the controls symmetry in the upper lip was restored, and the shape of the upper lip showed less variation after primary lip and soft palate closure. At this early age, retrusion of the soft-tissue mandible and chin, however, seems to be developing already.

### Funding
This work was supported by the Dutch Technology Foundation (Stichting Technische Wetenschappen) grant #10315 awarded to Thomas Maal. The funders had no role in study design, data collection and analysis, decision to publish, or preparation of the manuscript.

### Grant Disclosures
The following grant information was disclosed by the authors:
Dutch Technology Foundation (Stichting Technische Wetenschappen) grant #10315.

### Competing Interests
Anne Marie Kuijpers-Jagtman is an Academic Editor for PeerJ.

### Author Contributions
- Sander Brons conceived and designed the experiments, performed the experiments, analyzed the data, prepared figures and/or tables, approved the final draft.
- Jene W. Meulstee conceived and designed the experiments, performed the experiments, prepared figures and/or tables, authored or reviewed drafts of the paper, approved the final draft.
- Tom G.J. Loonen conceived and designed the experiments, performed the experiments, prepared figures and/or tables, authored or reviewed drafts of the paper, approved the final draft.
- Rania M. Nada authored or reviewed drafts of the paper, approved the final draft.
- Mette A.R. Kuijpers authored or reviewed drafts of the paper, approved the final draft.
- Ewald M. Bronkhorst analyzed the data, authored or reviewed drafts of the paper, approved the final draft.
- Stefaan J. Bergé conceived and designed the experiments, contributed reagents/materials/analysis tools, authored or reviewed drafts of the paper, approved the final draft.
- Thomas J.J. Maal conceived and designed the experiments, contributed reagents/materials/analysis tools, authored or reviewed drafts of the paper, approved the final draft.
- Anne Marie Kuijpers-Jagtman conceived and designed the experiments, analyzed the data, authored or reviewed drafts of the paper, approved the final draft.
## Human Ethics

The following information was supplied relating to ethical approvals (i.e., approving body and any reference numbers):

The study protocol was approved by the medical ethical commission of the Radboud University Medical Centre (Commissie Mensgebonden Onderzoek regio Arnhem-Nijmegen approval #2007/163).

## Data Availability

The raw data is available as Dataset S1.

## Supplemental Information

Supplemental information for this article can be found online at http://dx.doi.org/10.7717/peerj.7302#supplemental-information.

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
