# Peer review of "Three-dimensional facial development of children with unilateral cleft lip and palate during the first year of life in comparison with normative average faces"

_PeerJ, doi:10.7717/peerj.7302_

## Round 0.1 · original submission · Major Revisions

the paper would benefit from a major revision in which the issues raised should be addressed

·

Basic reporting

This was a prospective longitudinal 3D case-controlled morphometric study aimed at comparing 3D average faces of infants with unilateral cleft lip and palate with an age-matched control group at 3,6,9 and 12 months of age. The study is relevant and it enriches the body of published literature in this field. However, there are some areas in the manuscript that can be improved

Experimental design

Literature review
The present study was the first to compare average faces of infants with unilateral cleft lip and palate with age matched control group. However, there is another mixed longitudinal 3D case-controlled study examining facial morphometric asymmetry of infants with UCLP and UCL with age matched controls using stereophotogrammetric camera and Procrustes analysis. Their 3D facial morphology of cleft infants was captured before and after surgical repair, and during the period of early growth. It would be appropriate to add it to the literature review and compare their findings with your results in the discussion, especially that there are differences in the timing of the applied surgical protocols and the technique used for soft and hard palate repair;
1- Hood, C. A., Bock, M., Hosey, M. T., Bowman, A., & Ayoub, A. F. (2003). Facial asymmetry–3D assessment of infants with cleft lip & palate. International journal of paediatric dentistry, 13(6), 404-410.
2- https://ethos.bl.uk/OrderDetails.do?did=1&uin=uk.bl.ethos.432864

Methodology
3D image acquisition: Elaborate on the image acquisition procedure; was the distance between the infants and the camera standardized? capturing the whole face from ear to ear to generate one point cloud. The number of included vertices in each image. How many megapixel were in the texture map?
3D image processing
Procrustes transformation: elaborate on Procrustes transformation analysis
References
1- Line 435: change the letter “o” in PLoS to a capital “O”
2- In the reference “Mancini, L., Gibson, T. L., Grayson, B. H., Flores, R. L., Staffenberg, D., & Shetye, P. R. (2019) [line 436-439]; change the date of the publication from “Jan
1:1055665618771427” to “56(1), 31-38” to follow the used citation style

Validity of the findings

Construction of average faces provides an interesting perspective for diagnosis,
treatment planning, and comparison of facial shapes between controls and groups
with craniofacial deformities. This provides surgeons with valuable 3D templates in
planning the reconstruction of craniofacial anomalies. Orthodontists can use a soft
tissue reference for understanding facial soft tissue movement in relation to
dentoalveolar and skeletal changes occurring with treatment and growth. This might
develop to allow prediction of facial growth for sex and age group instead of relying
on 2D radiographic templates

Superimposing 3D faces provides a relevant tool in comparing and evaluating
surgical procedures outcome by superimposing pre and post-treatment facial images. Also, it can be employed in studying longitudinal soft tissue growth in cohort studies.

·

Basic reporting

The authors to be congratulated for this interesting piece of research which adds to our knowledge on the topic. The article is well written and the methodology is generally robust. The following are my concerns:
1. Methodology:
a. The lack of the errors of the method
the authors should provide an evidence of the reproducibility of the methodology, mainly the manual identification of the facial regions and the preauricular points used to identify the children's reference plane

b. It is not clear how the pose was standardised during capture? was the labial seal readily achievable in all the cases and how open lips were accounted for in the 3D facial analysis

2. Colour map

The authors used colour map to illustrate the morphological differences between UCLP cases and control, this lacks the directionality. The method dose not discriminate medial from lateral displacements not it separates upward from downward differences, these are essential to understated the pattern of dysmorphology between cleft and non-cleft cases.


3. Results:

a. I disagree that the asymmetry of the upper lip was completely restored, being 4 mm posterior to control. According to Figure 3, residual asymmetry is clear and appeared on the right side, the non-cleft side according to the authors, and this should be more anterior than posterior according to the colour coding of the image.

b. The authors explained at 12 months there is a remnant asymmetry on both sides of the nasal ridge, Figure 3 doesn't not confirm this findings. It is not clear how the asymmetry was assessed in this study ? and if it affected both sides then I would expect, a display of different colours, one blue and one red, of the nasal ridges which is not the case.


c. I am not sure of the authors interpretation of the morphological differences that were found between the two groups from 3-6 months of age as an indication of significant greater growth of the full face, nose, cheeks of UCLP cases. The nose is proven asymmetric according to figure 3, and cheeks are affected and displaced laterally on the cleft side, unfortunately the used analysis doesn't disclose the directionality (medio-laterally) of the measured differences. I also disagree with the statement of the significant more growth of the forehead in the UCLP cases. The axial rotation of the forehead along the axis of the facial reference used for detection of facial changes could lead to this findings.

d. Figure 3 doesn't support the claim that al all ages and mandible and chin were posterior to the control group. The captured images do not include the mandible! the ramus and body of the mandible were not included in the regions of interests! the blue colure at submental region is an area of prominence rather then deficiency according to the colour code.


4. Discussion
This section needs to be more focused and debate the two major findings of this study; the perceived greater facial growth at age 6 months of the UCLP case and the perceived mandibular deficiency at 12 months of the surgically manged cases. What is the biological bases of the overgrowth of the forehead in the UCLP cases?

Experimental design

see comments above

Validity of the findings

see comments above

Additional comments

see comment above

Reviewer 3 ·

Basic reporting

The authors presented their research clearly, with a sufficient field background review and literature references. The tables, figures are fine and results were well presented.

Experimental design

1. The authors aimed to investigate three-dimensional facial development of children with unilateral cleft lip and palate during the first year of life but the sample sizes were low as showed in the table 1 (13 in 3 months, 20 in 6 months, 16 in 9 months and 11 in 12 months) that the results could be biased due to the low sample sizes of the patient group. I suggest that the authors should increase the sample size of the patients.
2. The method of aligning 3D images based on Brons et al 2013 led to a biased data set towards the eye region, as indicated in line 214-215 "the least intersurface distance between UCLP and control facial morphology was noted in the region of the eyes".
3. Also because of the method of aligning 3D images, it was difficult to distinguish effects of facial growth and UCLP on the intersurface distance between UCLP and control facial morphology. I suggest the authors to clarify the ambiguity due to the aligning method applied.

Validity of the findings

The results of intersurface distance clearly showed the effects of surgical intervention of the cleft lip but the results failed to distinguish the effects of growth and UCLP. The differences between the UCLP and control could be originated from the intersurface difference in 3 months and the facial growth increased the difference up to 12 months; the same time the effects of UCLP on the difference were camouflage by the growth.

Additional comments

The major issue of the study is the low sample size of the patients. The other issue is the method of aligning 3D images based on Brons et al 2013 which led to a biased data set with least intersurface distance around the eye regions. Further the aligning method could not distinguish the effects on the facial growth and UCLP on the intersruface distance. The results showed clearly the effects of the surgical intervention on the upper lip but failed to provide effects of UCLP on the facial development. Minor issue is that the reference of Brons et al 2018 is not available yet.

---

## Round 0.2 · accepted · Accept

Your revision was favorably received by the reviewers.

·

Basic reporting

I would like to confirm that the authors had replied satisfactorily to the raised concerns. I would like to congratulate the authors for their seminal work and recommend accepting the manuscript for publication

Experimental design

The added sentences had met the reviewer expectations

Validity of the findings

The added sentences had met the reviewer expectations

Reviewer 3 ·

Basic reporting

Satisfied

Experimental design

The authors agreed with my previous comments and I have no further comments.

Validity of the findings

The authors agreed with my previous comments and I have no further comments.

Additional comments

Satisfied with revised version.